# Low Magnesium in Conjunction with High Homocysteine and Less Sleep Accelerates Telomere Attrition in Healthy Elderly Australian

**DOI:** 10.3390/ijms24020982

**Published:** 2023-01-04

**Authors:** Varinderpal S. Dhillon, Permal Deo, Philip Thomas, Michael Fenech

**Affiliations:** 1Health and Biomedical Innovation, Clinical and Health Sciences, University of South Australia, Adelaide, SA 5000, Australia; 2CSIRO Health and Biosecurity, Adelaide, SA 5001, Australia; 3Genome Health Foundation, North Brighton, SA 5048, Australia

**Keywords:** magnesium, sleep, homocysteine, telomere attrition

## Abstract

The relationship between sleep and micronutrients, including magnesium, is implicated in its regulation. The effects of low magnesium and other micronutrients on sleep disruption and telomere loss are not well understood. The present study was carried out in 172 healthy elderly subjects from South Australia. Plasma micronutrients including magnesium were measured. Each participant provided information about their sleep hours (<7 h or ≥7 h). Lymphocyte telomere length (TL) was measured by real-time qPCR assay. Plasma magnesium level was significantly low in subjects who sleep less than 7 h (*p* = 0.0002). TL was significantly shorter in people who are low in magnesium and sleep less than 7 h (*p* = 0.01). Plasma homocysteine (Hcy) is negatively associated with magnesium (r = −0.299; *p* < 0.0001). There is a significant interaction effect of magnesium and Hcy on sleep duration (*p* = 0.04) and TL (*p* = 0.003). Our results suggest that inadequate magnesium levels have an adverse impact on sleep and telomere attrition rate in cognitively normal elderly people, and this may be exacerbated by low levels of vitamin B_12_ and folate that elevate Hcy concentration.

## 1. Introduction

Telomere attrition is one of the most important and fundamental hallmarks of aging [1]. Telomere shortening or loss is increased when DNA breaks are induced by reactive oxygen species or as a result of DNA replication stress or defects in telomere maintenance [2,3].

Chronic inflammation and oxidative stress have been identified as pathogenic factors in aging and in age-related diseases and are partly caused by magnesium deficiency which results in excessive production of oxygen-derived free radicals (ROS) and low-grade inflammation [4,5]. Magnesium is an essential element required for the regulation of various cellular and metabolic reactions including ATP generation, DNA replication and DNA repair [6,7]. It also plays an important role in the folate–methionine–neurotransmitter cycle e.g., methionine adenosyltransferase and arylalkylamone-N-acetyletransferase require divalent Mg^2+^ as a co-factor [8] Magnesium is also required for the conversion of 5, 10-methylene tetrahydrofolate to 5-formyl tetrahydrofolate to be used in the neurotransmitter synthesis pathway (Figure 1) [9,10].

Telomeres are protective nucleoprotein structures at the ends of all chromosomes that provide genomic stability and prevent the loss of coding DNA. They are composed of a non-coding, repetitive DNA sequence (TTAGGG)n and associated shelterin proteins [11]. Telomeric sequences being guanine-rich are particularly sensitivity to ROS-induced oxidative damage [12]. The telomeric chromatin structure and integrity is impacted upon by Mg^2+^ biochemistry. The effect of magnesium on cellular aging may be related to its interactions with telomere homeostasis, telomere maintenance and activity of telomerase [6,13,14].

B-group vitamins B_2_, B_6_, B_9_ (folate) and B_12_, in their roles as substrates or co-factors in one-carbon metabolism, are required for methylation of Hcy to form methionine which is essential as a methyl donor for methylation reaction and for nucleotide synthesis and DNA repair [15,16]. The disruption of methylation processes leads to a build-up of Hcy thereby increasing the likelihood of inflammation, oxidative stress and subsequent damage to mitochondria and DNA [16,17]. The toxic effects of Hcy are at least partially mediated through oxidative damage to proteins and DNA [18,19]. An efficient detoxification of Hcy is therefore essential for maintaining genomic stability and cellular viability (Figure 1A cycle).

Sleep health is a multi-dimensional sleep-wakefulness pattern adapted to individual, social and environmental demands, which help promote physical and mental wellbeing [20]. Sleep deprivation affects glucose metabolism and neuroendocrine function [21]. Vitamin B_12_ contributes to melatonin secretion, and pyridoxine (vitamin B_6_) is involved in the synthesis of serotonin from tryptophan [22]. Folate and pyridoxine are involved in the conversion of tryptophan into serotonin (Figure 1B of cycle) [23].

Magnesium may also enhance melatonin secretion promoting sleep onset [22]. Magnesium is important for the synthesis of the N-acetyltransferase, which converts 5-hydroxytryptamine (5-HT) into N-acetyl-5-hydroxytryptamine, which can then be converted to melatonin [24]. Magnesium has also been found to be helpful in maintaining a normal circadian rhythm and sleep quality [25,26].

There are many reports in the literature that show inverse associations between sleep disturbances and telomere length [27,28,29,30,31]. However, there are no reports that investigated how magnesium deficiency affected sleep and telomere length, including how it (magnesium) affects the one-carbon cycle and its association with folate, B_12_ and Hcy, and how folate, B_12_ and Hcy affected sleep and telomere length. Therefore, the present study is designed to test the following hypotheses: (i) magnesium deficiency affects telomere length maintenance, (ii) magnesium deficiency interacts with folate and vitamin B_12_ to affect Hcy concentration and DNA integrity and (iii) magnesium deficiency affects melatonin synthesis and, as a consequence, sleep duration.

## 2. Results

### 2.1. Study Participants

The study cohort consisted of 136 females and 36 males. Mean age (yrs) and BMI (kg/m^2^) was 54.78 ± 1.2 and 27.32 ± 0.77 in men and 53.79 ± 0.71 and 26.49 ± 0.49 in women, respectively. Magnesium levels (mean) in plasma (mg/L) in men were 19.48 ± 0.21 and 19.32 ± 0.12 in women. Serum folate level (nm/L) in men was 33.78 ± 1.52 and 34.42 ± 0.85 in women, vitamin B_12_ level in plasma (pg/mL) in men was 414 ± 29.56 and 420.4 ± 18.21 in women, and Hcy levels (μmol/L) in men were 8.68 ± 0.44 and 7.95 ± 0.21 in women (Table 1). These characteristics including telomere length were not significantly different in male and female participants.

### 2.2. Relationship between Hcy, Serum Folate, Vitamin B_12_ and Magnesium

There was a significant inverse correlation between the concentrations of Hcy and serum folate (r = −0.310; *p* < 0.0001; Figure 2A), and vitamin B_12_ (r = −0.345; *p* < 0.0001; Figure 2A). However, the plasma magnesium concentration shows a positive correlation with folate (r = 0.236; *p* = 0.002; Figure 2B) and vitamin B_12_ (r = 0.204; *p* = 0.007; Figure 2B). Furthermore, plasma Hcy is negatively associated with magnesium (r = −0.299; *p* < 0.0001; Figure 2C). Serum folate concentration shows a positive but insignificant correlation with vitamin B_12_ (r = 0.10; *p* = 0.19; Figure 2D).

### 2.3. Correlation of TL with Serum Folate, Vitamin B_12_, Hcy and Magnesium

There was a significant negative correlation between TL and Hcy (r = −0.299; *p* < 0.0001; Figure 2E). However, magnesium shows a significant positive correlation with telomere length (r = 0.376; *p* < 0.0001 respectively; Figure 2F).

### 2.4. Association of Sleep with Hcy, Magnesium, Folate, Vitamin B_12_ and TL

Plasma Hcy was significantly higher in people who slept for less than 7 h compared to those who slept for 7 or more hours (*p* = 0.001; Figure 3A). The plasma magnesium level was significantly lower in those who slept less than 7 h compared to those who slept 7 or more hours (*p* = 0.0002; Figure 3B). Similarly, serum folate, vitamin B_12_ levels and TL (bp) were significantly lower in people who slept less than 7 h compared to those who slept 7 or more hours (*p* = 0.02; *p* = 0.009; *p* = 0.001; Figure 3C–E respectively). In bivariate analysis, plasma Hcy was not associated with age (r = 0.03; *p* = 0.69), BMI (r = −0.08; *p* = 0.32) or gender (*p* = 0.33), and magnesium was not found to be associated with age (r = 0.08; *p* = 0.29), BMI (r = 0.02; *p* = 0.84) or gender (*p* = 0.31) (Appendix A).

### 2.5. Effect of Serum Folate, Vitamin B_12_, Hcy, and Magnesium with Sleep Duration on TL

Our two-way ANOVA results indicate that people who had higher serum folate with adequate sleep duration (≥7 h or more) had significantly longer TL compared to those with low serum folate with less sleep duration (*p* < 0.0001; Figure 4A). TL was also significantly longer in people with higher serum folate levels compared to those with low serum folate who slept for less duration (*p* = 0.003; Figure 4A). TL was significantly shorter in people with low serum folate who slept for less duration compared to those with adequate sleep (*p* = 0.01; Figure 4A). TL was longer in people with high vitamin B12 and adequate sleep compared to those with low vitamin B_12_ and less sleep (*p* = 0.0002; Figure 4B). TL was also longer in people who had higher B_12_ levels irrespective of sleep duration (*p* = 0.02; Figure 4B). People with high Hcy and less sleep had significantly shorter TL compared to those with low Hcy and adequate sleep duration (*p* < 0.0001; Figure 4C). TL was also significantly shorter in people with high Hcy compared to those with low Hcy irrespective of sleep duration (*p* = 0.0002; Figure 4C) whereas TL was shorter in those with high Hcy and adequate sleep compared to those with low Hcy with less sleep duration (*p* = 0.0001; Figure 4C). TL was significantly longer in those with low Hcy irrespective of sleep duration (*p* = 0.03; Figure 4C). These observations indicate that folate, B_12_ and Hcy along with sleep duration significantly impacts TL in this cohort. We then investigated the effect of magnesium levels and sleep duration on TL. Our results indicate that those who had low magnesium levels and slept for less than 7 h had significantly shorter TL than those who slept 7 or more hours (*p* = 0.01; Figure 4D). TL was also significantly shorter in those who had high magnesium levels but slept less than 7 h than those who slept 7 or more hours (*p* = 0.03; Figure 4D). Similarly, TL was significantly longer in those who had slept less than 7 h but had high magnesium levels than those with low magnesium levels (*p* = 0.0003; Figure 4D). TL was also longer in those who had high magnesium levels in spite of adequate sleep duration (*p* < 0.0001; Figure 4D). These observations indicate that magnesium level in conjunction with sleep duration has a significant impact on TL.

### 2.6. Interaction Effects of Hcy and Magnesium on Sleep and TL

Our results show that there was a significant interaction of Hcy and magnesium levels with sleep duration (% interaction = 3.01; *p* = 0.04; Figure 5A). Similarly, there was significant interaction of Hcy and magnesium levels with TL (% interaction = 4.09; *p* = 0.003; Figure 5B).

Multiple regression analysis was also used to find out if any of these nutrients strongly predict the impact of magnesium on sleep and TL, taking into account the effect of any co-variates and interactive effects (Table 2). The multiple regression model examining the strength of blood biomarkers impacting on TL, were significantly pronounced in the actual cohort and sub-groups (*p* < 0.0001). This model indicates that Hcy and Mg had the strongest impact on TL.

## 3. Discussion

In the present study, plasma magnesium level was independently and positively associated with telomere length even after adjusting for covariates. The majority of previous reports suggest that magnesium might have a protective effect on telomere attrition rate [32,33,34] whereas O’Callaghan et al. [35] reported a negative association. It has also been reported that primary fibroblasts grown in a magnesium-deficient medium have increased telomere shortening and lose their replicative potential with accelerated expression of senescence-associated biomarkers [6]. It is highly likely that magnesium deficiency induced an increase in steady-state oxidative stress as is shown for many other micronutrient deficiencies [36]. Increased oxidative stress levels promote accelerated cellular senescence. If magnesium deficiency leads to increased oxidative stress, then telomeric DNA might be a sensitive target as some studies have shown that telomeres are prone to oxidant damage [6,37,38]. Here in this study, we have shown that people with low magnesium have shorter telomeres, thus indicating an increased telomere attrition rate. Magnesium is essential for a vast array of metabolic pathways and its levels like other micronutrients are constantly in flux. Therefore, homeostatic mechanisms must accommodate these changes in magnesium availability to maintain essential cellular functions such as ATP production. A consequence of magnesium deficiency is increased telomere attrition rate possibly due to DNA breaks induction, given that magnesium deficiency induces acentric chromosome deletions or fragments which lead to micronuclei formation and loss of this genetic material, including telomeric DNA loss from the main nuclei [39,40]. Although our cohort was healthy at the time of sampling, the shorter telomeres in people with low magnesium levels is not lethal, however, it can lead to accelerated tissue aging and make them more susceptible to oxidative damage leading to aging related diseases such as Alzheimer’s disease and cancers. Therefore, it is possible that magnesium plays an important role in telomere maintenance and its anti-inflammatory and anti-oxidative properties may impact on telomere length as shown in previous reports [41,42].

We also found that Hcy is strongly associated with magnesium. Recently it has been shown in pregnant women that treatment with magnesium sulfate and phentolamine resulted in the significant lowering of Hcy levels [43]. We can therefore anticipate that magnesium might play an important role in correcting Hcy levels. There are few reports in the literature that investigated the role of magnesium and Hcy together on TL. Higher Hcy is associated with shorter telomeres [17,43,44]. We then asked if magnesium level is associated with higher Hcy levels. We found that telomeres are significantly shorter in people who had low magnesium and high Hcy compared to those with high magnesium and low Hcy.

We have recently shown that inadequate sleep (<7 h) is associated with shorter TL [27]. We then investigated the association of magnesium with sleep. We found that magnesium levels are significantly higher in those who slept for 7 or more hours. Our results are in agreement with previous findings in animal models [45] and humans [46,47,48]. It is possible that magnesium may influence sleep duration via regulation of circadian rhythms [49]. Magnesium is important in neuronal processes as evident by a high correlation of a higher concentration of magnesium in the forebrain with sleep duration in rats [50]. In addition, magnesium is also associated with the synthesis of melatonin, a key hormone involved in the regulation of sleep–wake cycles [51]. It has been reported that rats who were fed a magnesium-deficient diet have decreased melatonin levels [45]. Sleep disturbances can cause neural impairment and are associated with neurodegenerative diseases and aging [52,53]. Various cellular homeostatic mechanisms that regulate the accumulation of toxic metabolites increased the cellular need for energy supplies and macromolecules; elevated neural damage and cellular stress play an important role in sleep [54,55,56,57,58].

Sleep deprivation can increase DNA damage and decrease the expression of DNA repair genes in human blood cells [52,59], thereby suggesting that sleep promotes nuclear maintenance i.e., balance between DNA damage and repair. It has been shown that stress-induced sleep is promoted by CEP-1, a member of the DNA damage response (DDR) pathway [60] that starts with the recruitment of poly(ADP-ribose) polymerase-1 (PARP-1) to the damage site. PARP-1is a detector of DNA damage and promotes the accumulation of downstream DDR proteins [61]. Zada et al. [62] showed dynamic clustering of DNA repair proteins under manipulation of sleep, neuronal activity and DNA damage and repair in transparent Zebra fish and revealed that DNA damage in neurons is a homeostatic driver of sleep and that PARP-1 can induce sleep that promotes chromosome dynamics and the recruitment and activity of DDR proteins. Therefore, it is plausible that sleep disturbances and/or less sleep can lead to reduced synthesis of PARP-1 which can then result in increased DNA damage including damage in telomeric DNA. PARP-1 is a sensor of DNA damage and is identified as a new melatonin dependent regulator of senescence-associated secretory phenotype (SASP) [63]. Melatonin is synthesized in the pineal body and it possesses a variety of physiological functions, including anti-oxidant, antiangiogenic, anti-inflammatory and sleep [64,65,66].

Poly(ADP-ribosyl)ation (PARylation) is a reversible process of post-translational modification of proteins and plays an important role in normal physiology as well as pathophysiological processes including regulation of DNA damage repair [61], chromatin structure [67] and transcription [68]. ADP-ribosylation is a reversible and site-specific post-transcriptional modification (PTM) that regulates cellular signaling pathways and its regulation is vital for the maintenance of genome integrity [69]. Its uncontrolled accumulation can lead to cell death. ADP-ribosyl acceptor hydrolase-3 (ARH-3) requires Mg^2+^ for its normal activity and if Mg is low or deficient, all these signaling pathways are affected, thereby leading to increased DNA damage and possibly telomere attrition.

## 4. Materials and Methods

### 4.1. Recruitment of Study Participants

Volunteers were recruited through (i) the Commonwealth Scientific and Industrial Research Organisation (CSIRO) Clinical Research Unit database in Adelaide, (ii) a local Channel 7 TV news report of this study and (iii) advertisements posted in hospitals and universities within the Adelaide metro area. A total of 172 healthy participants (35–65 years old) were recruited who passed the following inclusion criteria: non-smokers, not currently diagnosed with mild cognitive impairment (MCI) or AD, mini-mental state examination (MMSE) score ≥ 20, not on medication for life threatening diseases (e.g., chemotherapy), not taking daily minerals, fish oil and vitamin supplements above the Australian Recommended Dietary Allowance (RDA) level, able to understand the study protocol and not on cholesterol-lowering medication. The Human Ethics Committee of CSIRO approved the study. Participants were also asked to provide their sleep duration (<7 h or ≥7 h). Overnight fasted blood samples were collected at the CSIRO clinic by venepuncture. Lymphocytes were isolated from the fresh blood and stored at −80 °C until DNA was isolated to perform telomere length assay.

### 4.2. DNA Isolation and Real-Time qPCR Assay for Telomere Length (TL) Measurement

Blood was collected in EDTA tubes to isolate DNA. Genomic DNA was extracted from isolated lymphocytes using the QIAamp DNA blood mini kit (Qiagen, Melbourne, Australia). The OD ratio (1.8–2.0) of 260:280 was used an indicator of DNA purity. Any sample where OD ratio was not in an acceptable range was re-isolated before DNA quantification. Purified DNA samples were quantified using NanoDrop 1000 spectrophotometer (Thermo Fisher Scientific, Scoresby, Australia) and diluted as per experimental requirements (5 ng/μL). Telomere length was measured using quantitative real-time PCR as described previously [70]. The ratio of the telomere (T) repeat copy number to the single-copy gene (S) was determined for each sample using ABI 7300 Real-Time PCR Detection System (Life Technologies, Waltham, MA, USA). The final concentrations of the PCR reagents were 1 x SYBR Green Mix (Life Technologies, Waltham, MA, USA), 20 ng DNA, 0.2 μmol of telomere specific primers (F: 5′-GGTTTTTGAGGGTGAGGGTGAGGGTGAGGGTAGGGT-3′; R: 5′-TCCCGACT ATCCCTATCCCTATCCCTATCCCTATCCCTA-3′) and 0.3 μmol of 36B4 primers (F: 5′-CAGCAAGTGGGAAGGTGTAATCC-3′; R: 5′-CCCATTCTATCATCAA CGGGTCAA-3′). The reactions were performed using telomere and 36B4 specific primers in a 96-well plate, and each plate included a reference DNA sample. It may be noted that primers 36B4 can amplify pseudogene regions on chromosomes 1, 2, 5, 12 and 18. A five-point serial dilution standard curve of DNA concentration versus T/S ratio using DNA isolated from the 1301 cell line (which has a mean telomere length of 23,000 base pairs) was established in each plate. The standard curve was then used to convert the T/S ratio into telomere length (TL) in base pairs (bp) using the following equation: Absolute TL(bp) = 2433.23X + 3109.51 where X = T/S ratio, 2433.23 is the slope and 3109.51 is the intercept of the standard curve [71]. To generate the standard curve and convert T/S ratio into absolute TL (bp), DNA isolated from IMR90 cell line was used. DNA was isolated from cells taken from cultured cell line at different population doubling time (~35 h; 10 time points). Southern blot method (TRF) was used to determine TL of these DNAs isolated from 10 different time points of sub-culturing. DNA isolated from these cells was also used in qPCR assay to determine T/S ratios as a part to generate the initial standard curve which was then converted into base pairs. The formula generated from this was later on used to convert T/S ratios to absolute telomere length: absolute TL(bp) = 2433.23X + 3109.51 where X = T/S ratio of samples, 2433.23 is the slope and 3109.51 is the intercept [72]. A standard curve with a high correlation factor (R^2^ ≥ 0.97) was required to accept the results from the plate. The intra-assay coefficient between triplicates was 2.9% for telomeres and 1.7% for the single-copy gene, whereas the inter-assay CV between plates was 0.6% for telomeres and 0.73% for the single-copy gene.

### 4.3. Micronutrient Analyses

Blood was collected in 2 mL serum tubes and kept at room temperature for 30–45 min before being processed by SA Pathology for serum folate analysis. Blood was collected in lithium heparin tubes for plasma homocysteine and vitamin B12 analysis. Lithium heparin tubes containing blood were transported to SA pathology on ice. Serum and plasma were separated and analysed on the same day they were collected as per standard protocols by SA Pathology. Serum folate, plasma homocysteine and vitamin B_12_ concentrations was measured by an Architect^®^ analyser (Abbott Laboratories, Abbott Park, IL, USA) in the department of Chemical pathology certified diagnostic laboratory of SA Pathology (Adelaide, Australia).

For magnesium analysis, plasma was isolated from blood collected in a lithium heparin tube and stored at −80 °C before analysis. The concentration of total plasma magnesium (Mg) was measured by ICP-MS by Waite Analytical Services, Adelaide, South Australia. The coefficient of variation of duplicate measurements did not exceed 5%.

### 4.4. Statistical Analysis

Parametric statistical methods were used for biomarkers exhibiting Gaussian distribution. Non-parametric methods were employed to analyse the results for biomarkers that were not Gaussian in their distribution. Correlation analysis was performed by Spearman’s or Pearson’s test depending on whether the biomarker data were Gaussian or non-Gaussian in their distribution. Descriptive statistics were used to summarize demographic characteristics. We also performed a two-way ANOVA to measure the interactive effects of two factors on a specific biomarker (e.g., effect of Hcy and Mg on hours of sleep, and effects of Hcy and Mg on telomere length). Statistical tests were performed using GraphPad Prism (Version 9.0; San Diego, CA, USA) and SPSS (IBM SPSS Version 23; Sydney, Australia). Significance for all statistical tests was set at *p* < 0.05 for all analyses.

## 5. Conclusions

Overall, it can be concluded that low magnesium levels can influence sleep on one hand and accelerates telomere attrition on the other hand. Furthermore, less sleep can also accelerate telomere attrition in healthy elderly people (Figure 6). Similarly, other micronutrients such as low folate, vitamin B_12_ and high homocysteine in association with inadequate sleep duration also impacts on telomere length, and their interaction with magnesium deficiency may further aggravate cellular DNA damage and accelerate senescence.

In conclusion, the results from our study support the hypothesis that the optimal intake of micronutrients such as magnesium, folate and vitamin B_12_ is essential for telomere maintenance and that such effects may be further mediated by impaired sleep duration. Whether improved sleep duration and quality may independently prevent telomere attrition should be verified in intervention studies that are controlled for magnesium and B vitamin status.

## Figures and Tables

**Figure 1 ijms-24-00982-f001:**
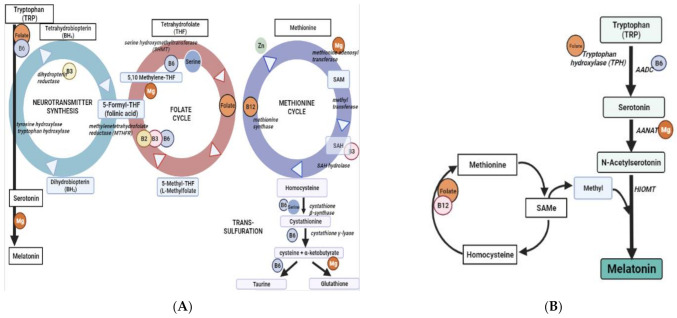
(**A**) Folate–methionine pathway and its interaction with neurotransmitter synthesis (melatonin) pathway, and (**B**) detailed interactive pathway of SAMe synthesis and donation of methyl group by SAMe for melatonin synthesis.

**Figure 2 ijms-24-00982-f002:**
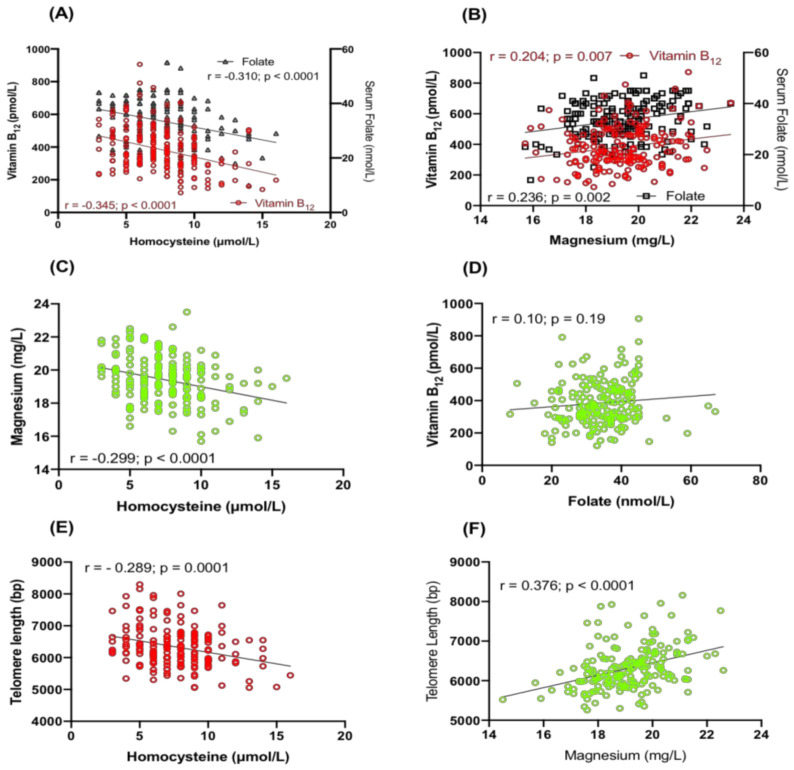
(**A**) The correlation of vitamin B_12_ (left panel) and serum folate (right panel) with homocysteine; (**B**) correlation of vitamin B_12_ (left panel) and serum folate (right panel) with magnesium; (**C**) correlation of magnesium with homocysteine; (**D**) correlation of vitamin B_12_ with serum folate; (**E**) correlation of telomere length with homocysteine, and (**F**) correlation of telomere length with magnesium.

**Figure 3 ijms-24-00982-f003:**
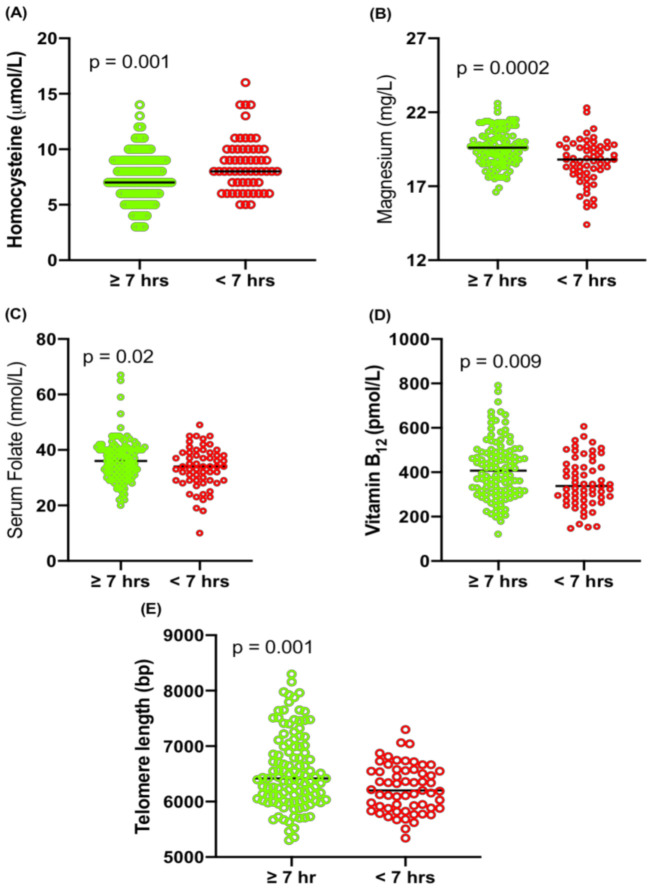
(**A**) Homocysteine concentration in subjects who slept ≥7 h (n = 113; Male 22; Female 91) and <7 h (n = 59; Male 14; Female 45); (**B**) magnesium levels in subjects who slept for ≥7 h and <7 h; (**C**) serum folate concentration in subjects who slept ≥7 h and <7 h; (**D**) vitamin B_12_ concentration in subjects who slept ≥7 h and <7 h and (**E**) telomere length (bp) in subjects who slept for ≥7 h and <7 h.

**Figure 4 ijms-24-00982-f004:**
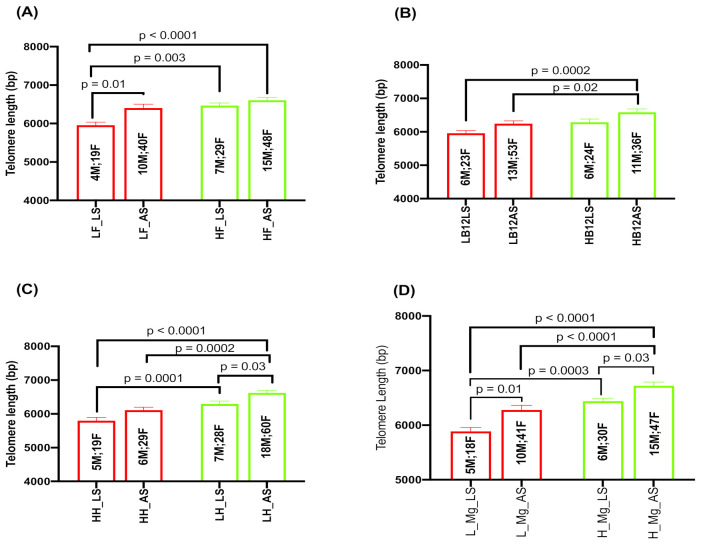
(**A**) Association of sleep duration (LS: <7 h; AS: ≥7 h) with telomere length in subjects with low folate (LF) and high folate (HF); (**B**) association of sleep duration with TL in subjects with low B_12_ (LB12) and high B_12_ (HB12); (**C**) association of sleep duration with TL in subjects with low homocysteine (LH) and high homocysteine (HH) and (**D**) association of sleep duration with TL in subjects with low magnesium (L Mg) and high magnesium (H Mg). Number of subjects in each category (male and female) is shown in the middle of each bar.

**Figure 5 ijms-24-00982-f005:**
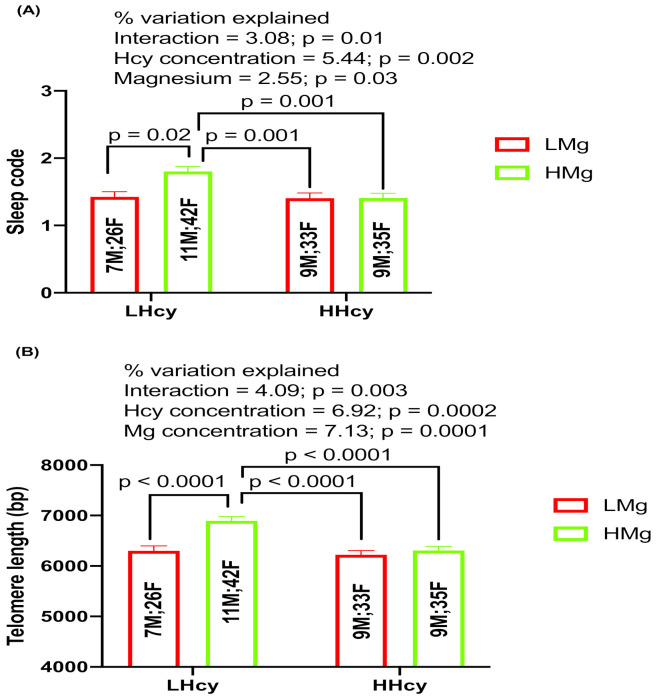
(**A**) Two-way ANOVA analysis of independent and interactive effects of homocysteine and magnesium on sleep duration (sleep code as 1 for <7 h and sleep code 2 as ≥7 h), and (**B**) two-way ANOVA analysis of independent and interactive effects of homocysteine and magnesium on telomere length. Number of subjects in each category (male and female) is shown in the middle of each bar.

**Figure 6 ijms-24-00982-f006:**
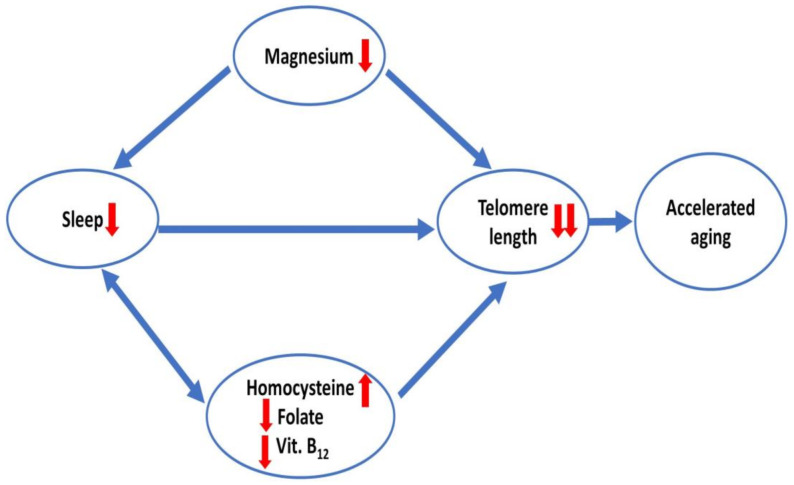
Magnesium as a mediator of the effects of sleep and homocysteine (in association with folate and vitamin B_12_) on telomere length leading to accelerated aging.

**Table 1 ijms-24-00982-t001:** Baseline characteristics of the study participants.

	Male	Female
Total number (n =172)	36	136
Mean age (years)	54.78 ± 1.2	53.79 ± 0.71
BMI (kg/m^2^)	27.32 ± 0.77	26.49 ± 0.49
Magnesium (mg/L)	19.48 ± 0.21	19.32 ± 0.12
Serum folate (nm/L)	33.78 ± 1.52	34.42 ± 0.85
Vitamin B_12_ (pg/mL)	414.3 ± 29.56	420.4 ± 18.21
Homocysteine (μmol/L)	8.98 ± 0.44	8.65 ± 0.21
Telomere length (bp)	6328 ± 56.76	6425 ± 60.66
Exercise		
At least once a week	27	100
Seldom/Never	9	36
Working Hours		
Less than 9 h	32	121
9 h or more	4	15
Stress		
Light/Moderate	32	119
Excessive	4	17
Breakfast		
Every morning	33	128
Often skip	3	8
Nutritional habit (3–4 serves of fruit/vegetable)		
Good/Moderate	34	127
Poor	2	9

**Table 2 ijms-24-00982-t002:** Multiple regression analysis in the total cohort and sub-groups for the association of TL in peripheral blood lymphocytes with various independent variables.

Independent Variable	Total Cohort	Male	Female	<7 h Sleep	≥7 h Sleep
Gender	0.113	-	-	0.121	0.11
Age	−0.219	−0.216	−0.229	−0.212	−0.224
BMI	−0.015	−0.009	−0.018	−0.021	−0.013
Folate	0.143	0.147	0.142	0.133	0.149
B_12_	0.104	0.11	0.102	0.095	0.107
Hcy	−0.312 *	−0.307 *	−0.314 *	−0.397 *	−0.296
Mg	0.298 *	0.291 *	0.299 *	0.243 *	0.342
Multiple R					
	0.604 *	0.597 *	0.605 *	0.433	0.463

*: significant at *p* < 0.0001.

## Data Availability

Data will be uploaded to a publicly available repository upon acceptance of the manuscript.

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
