# Peer review of "Low Magnesium in Conjunction with High Homocysteine and Less Sleep Accelerates Telomere Attrition in Healthy Elderly Australian"

_ijms, 2023, doi:10.3390/ijms24020982_

Round 1

Reviewer 1 Report

This is a very interesting hypothesis-driven study searching how low magnesium and other micronutrients impact on sleep disruption and telomere attrition, in which the rationale of the hypothesis is well-formulated thanks also to the presence of a figure who summarizes it well. The paper is written clearly, methods are quite comprehensively well described. Results are well presented using figures and tables.

Characteristics of the population should, however, be improved. There are, in fact, poor descriptions of confounders info such as occupations and lifestyle, apart from sleep duration, that includes information on jobs, sedentary or stressful conditions, nutritional habits of participants, physical activity, and/or other environmental factors that have possible effects on telomere length(TL). Is there any information on cells' blood and hematochemical parameters?

Furthermore, I think that additional multivariate statistical analysis should be performed- eg, multivariate analysis or age-adjusted mean, in order to take into account the effect, in particular of age or other confounders, on the simple regression analysis and the ANOVA comparisons (figures 2-5).

In Discussion the first sentence tells:” In the present study, plasma magnesium level was independently and positively associated with telomere length even after adjusting for covariates.” Which are the covariates taken into consideration? There is no description of those in the materials and methods section.

Author Response

  1. Characteristics of the population should, however, be improved. There are, in fact, poor descriptions of confounders info such as occupations and lifestyle, apart from sleep duration, that includes information on jobs, sedentary or stressful conditions, nutritional habits of participants, physical activity, and/or other environmental factors that have possible effects on telomere length (TL). Is there any information on cells' blood and hematochemical parameters?

Response

We have added more information in Table 1 regarding the lifestyle factors e.g. exercise, working hours, stress, breakfast and nutritional habits obtained using the Health Promotion Index tool. While collecting the information at the recruitment time, we did not ask for any environmental factors that can impact on TL. We did not collect data regarding the blood/hematochemical parameters.

  1. Furthermore, I think that additional multivariate statistical analysis should be performed- eg, multivariate analysis or age-adjusted mean, in order to take into account the effect, in particular of age or other confounders, on the simple regression analysis and the ANOVA comparisons (figures 2-5).

Response

We have added an additional Table (2) in the revised manuscript about the multiple regression analysis in the total cohort and as well as sub-groups. The results were adjusted for age, BMI and gender (the known confounders that can impact on TL). Subsequently, we have added a paragraph in the revised manuscript. The following information is added at the end of the results:

“Multiple regression analysis was also used to find out if any of these nutrients strongly predict the impact of magnesium on sleep and TL, taking into account the effect of any co-variates and interactive effects (Table 2). The multiple regression model examining the strength of blood biomarkers impacting on TL were significantly pronounced in the actual cohort and sub-groups (p < 0.0001). This model indicates that Hcy and Mg had the strongest impact on TL.”

Reviewer 2 Report

The authors of the study showed that low magnesium levels combined with high homocysteine and less sleep accelerated telomere shortening in healthy older Australians. Research reports that many processes affect telomere length. Lack of sleep, chronic inflammation or cardiovascular diseases affect the shortening of telomere length, in contrast to a healthy diet, physical activity and lack of stress, which positively affect the length of telomeres. The results obtained in the work were predictable and are not new. The conclusions of the work are supported by the results. The work contains figures and diagrams. The work may be published in a journal.

Author Response

Attached please find the attached document 

Reviewer 3 Report

The article by Dhillon et al., reads well with some interesting findings.  But concerns were raised that need addressing.

Major Comments

1. A major concern focuses on spurious effects being detected with the sample sizes being analyzed in Figure 3A-E; Figure 4A-D; and Figure 5A-B .  There are 136 females and 36 males which are well matched but no mention of the breakdown of these participants when looking at plasma homocysteine, plasma magnesium, serum folate, plasma vitamin B12 and telomere length.   To rule out any spurious findings, suggest adding the sample number of M:F to each group being analyzed in Figures 3A-E; 4A-D and 5A-B. Adding any power estimates would also enhance this analysis too.

2. How do you justify not seeing spurious effects with the statement on Line 131-133 that says "TL was also significantly longer in people with higher serum folate levels compared to those with low serum folate who slept for less duration"" when Figure 4A pictures similar telomere length values between the low folate >7hrs sleep and high folate <7hrs sleep?  

3. Similarly, Figure 4B has similar TL pictured for the low Vitamin B12_ >7hrs sleep and high Vitamin B12 <7hrs sleep.  How can you justify the sentences on Lines 133-138 that says TL was longer in people with high vitamin B12 and adequate sleep as well as TL was also longer in people who had higher B12 levels irrespective of sleep duration when neither of these hold for these two groups with similar TL?  Please justify these findings with the sample numbers involved in this analysis to rule out spurious effects.

4. Plasma homocysteine, plasma magnesium and plasma Vitamin B12 were investigated in this study but serum folate was also analyzed. In the Abstract it states plasma micronutrients were measured.  How was serum folate analyzed when plasma was collected? Please include the details of how the serum was collection to measure folate and how plasma was collected from the venepuncture in the Materials and Methods section too.

5. In the Materials and Methods section, purified DNA samples were mentioned to be quantified but there is no mention of how the DNA was purified.  Please include this description.

6. Telomere length was conducted using the Cawthon (2002) relative TL qPCR assay.  The 36B4 primers were selected to measure the single copy gene (S). Since the 36B4 primers are well known to amplify up other 75bp pseudogene regions on chromosomes 12 and 2 as well as 18, 5 and 1 (DOI: 10.1007/978-1-4939-8931-7_5), please state the limitations for using the 36B4 primers in this study.

7. Please also explain how the 23,000 base pair length was determined for the 1301 cell line?  Other papers find longer telomeres when using this 1301 cell line (O'Callaghan et al, 2008, DOI: 10.2144/000112761 found 70kb length). Please list the advantages in the paper for using the 1301 cell line for the standard curve?

8. The reference given [68] did not explain the rationale behind converting the T/S ratio using the standard curve and the following equation: Absolute TL(bp) = 2433.23X + 3109.51 where X = T/S ratio, 2433.23 is the slope and 3109.51 is the intercept of the standard curve. Do these values change if the slope and intercept change with each standard curve? Why not use manufactured oligomers of known length to calculate telomere length in base pair values as per the O'Callaghan et al, 2008 study (DOI: 10.2144/000112761)?

9. Lines 119-122: Results from bivariate analysis is mentioned but it is not clear if this is related to results shown in Figure 3A-F as age and BMI are also a factor in this analysis. Are these groups shown in the paper before this analysis? Are they from Table 1?  Please explain.

Minor Comments

1. Figure 1 shows a graphical representation of the role magnesium plays in the 5,10-methylene tetrahydrofolate conversion but a reference is needed to support the information contained in Figure 1.

2. Reference #10 on Line 48 is used to support the statement "telomere sequences as guanine rich being sensitive to ROS-induced oxidative damage."  Reference #10 by Neeley and Essigmann does not mention telomeres or ROS oxidative damage at all. Please ensure correct use of this reference and replace with the correct one.

3. Line 49: Missing full stop at the end of "Telomeric chromatin structure and integrity is impacted upon by Mg2+biochemistry"

4. In the introduction and Figure 6 in the conclusion, B9  or B9(folate) is mentioned but in the results and materials and methods, only folate is used.  Please be consistent in naming this vitamin to avoid confusion.

Author Response

Reviewer #3

  1. A major concern focuses on spurious effects being detected with the sample sizes being analyzed in Figure 3A-E; Figure 4A-D; and Figure 5A-B. There are 136 females and 36 males which are well matched but no mention of the breakdown of these participants when looking at plasma homocysteine, plasma magnesium, serum folate, plasma vitamin B12 and telomere length.   To rule out any spurious findings, suggest adding the sample number of M:F to each group being analyzed in Figures 3A-E; 4A-D and 5A-B. Adding any power estimates would also enhance this analysis too.

Response

We have added the number of subjects (male and female) in Figures 3-5. Power estimates were not performed as this does not improve the significance of the results obtained. The large number of significant P values in many of the relevant comparisons we made is indicative that the study was adequately powered. Nevetheless, we acknowledge that a larger number of subjects may be required to further verify the trends observed.  

  1. How do you justify not seeing spurious effects with the statement on Line 131-133 that says "TL was also significantly longer in people with higher serum folate levels compared to those with low serum folate who slept for less duration"" when Figure 4A pictures similar telomere length values between the low folate >7hrs sleep and high folate <7hrs sleep?  

Response

We justify the statement "TL was also significantly longer in people with higher serum folate levels compared to those with low serum folate who slept for less duration" because the detrimental effects of <7hrs sleep on TL are negated by the beneficial effects of more folate on TL and vice-versa. In other words, the detrimental effects of less sleep on TL tend to be more evident in those with low folate status.

  1. Similarly, Figure 4B has similar TL pictured for the low Vitamin B12_ >7hrs sleep and high Vitamin B12 <7hrs sleep.  How can you justify the sentences on Lines 133-138 that says TL was longer in people with high vitamin B12and adequate sleep as well as TL was also longer in people who had higher B12levels irrespective of sleep duration when neither of these hold for these two groups with similar TL?  Please justify these findings with the sample numbers involved in this analysis to rule out spurious effects.

Response

Similarly, we justify the statement " TL was longer in people with high vitamin B12 and adequate sleep compared to those with low vitamin B12 and less sleep (p = 0.0002; Figure 4B). TL was also longer in people who had higher B12 levels irrespective of sleep duration (p = 0.02; Figure 4B)" because the detrimental effects of <7hrs sleep on TL are negated by the beneficial effects of more B12 on TL and vice-versa. In other words, the detrimental effects of less sleep on TL tend to become more evident in those with low B12 status. Furthermore, these trends are consistent with those for folate and homocysteine (the latter being an inverse metabolic biomarker of folate and/or vitamin B12 status) all of which are key players in one-carbon metabolism, and therefore are also biologically plausible given that folate and/or vitamin B12 deficiency can result in DNA breaks and loss of telomeres (Bull et al., 2014; Praveen et al., 2022). Based on these arguments we are confident that the observed trends are not spurious effects.    

Bull CF, Mayrhofer G, O'Callaghan NJ, Au AY, Pickett HA, Low GK, Zeegers D, Hande MP, Fenech MF (2014) Folate deficiency induces dysfunctional long and short telomeres; both states are associated with hypomethylation and DNA damage in human WIL2-NS cells. Cancer Prev Res (Phila) 7:128-138.

Praveen G, Sivaprasad M, Reddy GB (2022) Telomere length and vitamin B12. Vitam Horm 119:299-324.

  1. Plasma homocysteine, plasma magnesium and plasma Vitamin B12were investigated in this study but serum folate was also analyzed. In the Abstract it states plasma micronutrients were measured.  How was serum folate analyzed when plasma was collected? Please include the details of how the serum was collection to measure folate and how plasma was collected from the venepuncture in the Materials and Methods section too.

Response

Blood was collected in 2 mL serum tubes and kept at room temperature for 30-45 minutes before being processed by SA Pathology for serum folate analysis. Blood was collected in lithium heparin tubes for plasma homocysteine and vitamin B12 analysis. Lithium heparin tubes containing blood were transported to SA pathology on ice. Serum and plasma were separated and analysed on the same day they were collected as per standard protocols by SA Pathology.

Plasma was isolated from blood collected in lithium heparin tube for magnesium analysis and stored at -80°C before analysis.

  1. In the Materials and Methods section, purified DNA samples were mentioned to be quantified but there is no mention of how the DNA was purified.  Please include this description.

Response

OD ratio (1.8-2.0) of 260:280 was used an indicator of DNA purity. Any sample where OD ratio was not in acceptable range, was re-isolated before DNA quantification. This information has been added in the relevant section of the revised manuscript.

  1. Telomere length was conducted using the Cawthon (2002) relative TL qPCR assay.  The 36B4 primers were selected to measure the single copy gene (S). Since the 36B4 primers are well known to amplify up other 75bp pseudogene regions on chromosomes 12 and 2 as well as 18, 5 and 1 (DOI: 10.1007/978-1-4939-8931-7_5), please state the limitations for using the 36B4 primers in this study.

Response

We are aware that primers 36B4 used in the present study can amplify pseudogene regions on chromosomes 1, 2, 5, 12 and 18. However, we assume that it amplifies these regions in all subjects and therefore, it does not interfere in the overall results. We have mentioned the limitation of using 36B4 primers in the relevant section.

  1. Please also explain how the 23,000 base pair length was determined for the 1301 cell line?  Other papers find longer telomeres when using this 1301 cell line (O'Callaghan et al, 2008, DOI: 10.2144/000112761 found 70kb length). Please list the advantages in the paper for using the 1301 cell line for the standard curve?

Response

There is a lot of variation reported in the literature about the TL of 1301 cell line (Hultdin et al., 1998 and Jeyapalan et al., 2006; 25-80Kbp respectively in addition to mentioned in O’Callaghan et al., 2008; 70Kbp). This cell line (1301) is a tetraploid cell line with long telomeres. There is evidence that further sub-culturing of this cell line may lead to lengthening of telomeres (Hultdin et al., 1998), which is perhaps the reason for this significant variation in its reported TL. Therefore, we sent DNA isolated from the cell line to a laboratory that performs Southern blot (TRF) method to measure TL. The results obtained using Southern blot method indicated that the mean TL of the 1301 cell line we used is 23 Kb. DNA isolated from the lot used in Southern Blot analysis was used as a positive control in all our experiments.

Hultdin M, Gronlund E, Norrback K-F, Eriksson-Lindstrom E, Just T, Roos G (1998) Telomere analysis by fluorescence in situ hybridization and flow cytometry. Nucleic Acid Res 26:3651-3656.

Jeyapalan JC, Saretzki G, Leake A, Tilby MJ, von Zglinicki T (2006) Tumour-cell apoptosis after cisplatin treatment is not telomere dependent. Int J Cancer 118:2727-2734.

  1. The reference given [68] did not explain the rationale behind converting the T/S ratio using the standard curve and the following equation: Absolute TL(bp) = 2433.23X + 3109.51 where X = T/S ratio, 2433.23 is the slope and 3109.51 is the intercept of the standard curve. Do these values change if the slope and intercept change with each standard curve? Why not use manufactured oligomers of known length to calculate telomere length in base pair values as per the O'Callaghan et al, 2008 study (DOI: 10.2144/000112761)?

Response

The slope and intercept of the equation used to convert T/S ratio to absolute TL was determined from six different experiments using the 1301 cell line. A five point serial dilution was set up in six different rows on one plate and this experiment was repeated six times. The slope and intercept of the equation TL (bp) = 2433.23X + 3109.51 is the mean of these parameters deduced from these experiments. Every plate has been run with these standards and a correlation factor of R2 ³ 0.97 was required to accept the results.

Oligomer (84b) mentioned in our previous report (O’Callaghan et al., 2008) can be used. However, with our further knowledge, we suspect that this oligomer might form G-quadruplex when reconstituted for further serial dilutions. G-quadruplex secondary structures are formed in the guanine rich regions (TTAGGG)n of the telomeres and this oligomer is 84 bases long sequence (TTAGGG)14. Furthermore, the results obtained using this short synthetic oligomer have been inconsistent in our experience and may be biased with respect to G-quadruplex formation which may alter DNA replication efficiency (Bryan, 2020) and as a consequence influence the TL assay results . Therefore, we decided not to use this synthetic oligomer in our recent and further studies.

Bryan TM (2020) G-Quadruplexes at Telomeres: Friend or Foe? Molecules 25:3686.

  1. Lines 119-122: Results from bivariate analysis is mentioned but it is not clear if this is related to results shown in Figure 3A-F as age and BMI are also a factor in this analysis. Are these groups shown in the paper before this analysis? Are they from Table 1?  Please explain.

Response

This analysis is not related to Figure 3A-F. The results are not significant and therefore, we did not provide the figure. However, to avoid any confusion, we have added a new supplementary figure for these results.

Minor comments

  1. Figure 1 shows a graphical representation of the role magnesium plays in the 5,10-methylene tetrahydrofolate conversion but a reference is needed to support the information contained in Figure 1.

Response: The following two new references has been added:

Christensen KE, Mirza IA, Berghuis AM, MacEenzie RE (2005) Magnesium and phosphate ions enable NAD binding to methylenetetrahydrofolate dehydrogenase-methenyltetrahydrofolate cyclohydrolase. J Biol Chemistry 280:34316-34323.

Zhao LN, Kaldis P (2022) The catalytic mechanism of the mitochondrial methylenetetrahydrofolate dehydrogenase/cyclohydrolase. PLoS Comput Biol 18:e1010140.

  1. Reference #10 on Line 48 is used to support the statement "telomere sequences as guanine rich being sensitive to ROS-induced oxidative damage."  Reference #10 by Neeley and Essigmann does not mention telomeres or ROS oxidative damage at all. Please ensure correct use of this reference and replace with the correct one.

Response

We have corrected the reference as follows:

Fouquerel E, Barnes RP, Uttan S, Watkins SC, Bruchez MP, Opresko Pl (2019) Targeted and persistent 8-oxoguanine base damage at telomeres promotes telomere loss and crisis. Molecular Cell 75:117-130.

  1. Line 49: Missing full stop at the end of "Telomeric chromatin structure and integrity is impacted upon by Mg2+biochemistry"

Response: corrected

  1. In the introduction and Figure 6 in the conclusion, B9 or B9 (folate) is mentioned but in the results and materials and methods, only folate is used.  Please be consistent in naming this vitamin to avoid confusion.

Response

We have now used ‘folate’ consistently throughout the revised manuscript.

Round 2

Reviewer 1 Report

Authors answered all the posed questions. 

Author Response

We thank the Reviewer for valuable comments in Revision 1

Reviewer 3 Report

I thank the authors for their reply to my comments.  Upon reading the response, I have additional questions arising from this updated information. They are listed below:

1. If blood was collected in lithium heparin tubes for serum and plasma collection, please include the type of vacutainer used to collect blood for DNA extraction from lymphocytes. 

2. If heparin tubes were used to collect blood for DNA extraction and since heparin should be avoided as it can bind to DNA during purification and inhibit TaqPolymerase in the PCR amplification steps (doi: 10.22074/cellj.2015.526), please include for each telomeric repeat and 36B4 qPCR assay:

a) the number of sample replicates used

b) the percentage of DNA samples repeated and failing quality control (QC)

c) the number of samples excluded from further analysis 

d) the method for accounting for variation between replicates 

e) the method for accounting for well position effects within and between plates.

Kotikalapudi R, Patel RK (2015), Comparative study of the influence of EDTA and sodium heparin on long term storage of cattle DNA. Cell J;17(1):181-6.

3. Lines 287-290 states "OD ratio (1.8-2.0) of 260:280 was used an indicator of DNA purity. Any sample where OD ratio was not in acceptable range, was re-isolated before DNA quantification. This information has been added in the relevant section of the revised manuscript." I assume the OD ratio was performed on the genomic DNA that was already extracted. Please include how many blood samples were needing to be thawed for DNA to be re-extracted again?

4. I thank the authors for answering the question about using their standard curve derived from the 1301 cell line DNA to convert the T/S ratio into telomere length in base pair (bp) values. However, this method does not provide absolute base pair lengths in bp values ; but rather mean relative telomere length measures. Please include how this method measures absolute telomere lengths in base pair values. 

5. If using DNA extracted from the 1301 cell line of known length as the positive control for the standard curve, please include what measures are in place for storing this DNA to avoid telomere length attrition/degradation?

5. Lines 305-308 states "The standard curve was then used to convert the T/S ratio into telomere length (TL) in base pairs (bp) using the following equation: Absolute 306 TL(bp) = 2433.23X + 3109.51 where X = T/S ratio, 2433.23 is the slope and 3109.51 is the intercept of the standard curve [71]." This reference does not mentioned using telomere length nor this equation. Please update this reference.

6. I also thank the authors for including the M:F sample numbers in the figures.  I believe the only difference in Figure 5 is on the Y axis with the sleep code (Figure 5A) and the telomere length (Figure 5B) however I notice a substantial difference in the M:F  on the X axis which still add up to 136 females and 36 males.  Please include why some samples jump from being L-Hcy to H-Hcy or being L Mg to H Mg when measuring telomere length from sleep code as shown in Figures 5A and 5B?  This may account for some of the statistical significance found in these two figures.

7. Similarly, please include how Figure 4c now has 144 females and 36 males listed in this figure? 

Author Response

Reviewer 3

I thank the authors for their reply to my comments.  Upon reading the response, I have additional questions arising from this updated information. They are listed below:

  1. If blood was collected in lithium heparin tubes for serum and plasma collection, please include the type of vacutainer used to collect blood for DNA extraction from lymphocytes.

Response: Blood was collected EDTA tube for DNA related assays. (We used appropriate tubes to collect blood for various assays). We have added the following information in the methods section: “Blood was collected in EDTA tube to isolate DNA”.

  1. If heparin tubes were used to collect blood for DNA extraction and since heparin should be avoided as it can bind to DNA during purification and inhibit TaqPolymerase in the PCR amplification steps (doi: 10.22074/cellj.2015.526), please include for each telomeric repeat and 36B4 qPCR assay:

  1. a) the number of sample replicates used

Response: Three replicates from each sample were used to calculate T/S ratio.

  1. b) the percentage of DNA samples repeated and failing quality control (QC)

Response: Approximately 2% samples failed the quality control as CT values obtained were not within acceptable range. These samples were re-run to be included in the further analysis.

  1. c) the number of samples excluded from further analysis

Response: No sample was excluded from the final analysis.

  1. d) the method for accounting for variation between replicates

Response: If CT values within triplicate differs by 0.5. The results were accepted. Any sample where CT values using telomere or 36B4 primers was not within this acceptable range was discarded and re-run to be included in the final assay. Mean CT value was calculated from these triplicates. However, sample(s) where CT from two replicates is within the acceptable range was accepted to calculate mean CT value.

  1. e) the method for accounting for well position effects within and between plates.

Response: As mentioned in our previous response, six rows and six different experiments were used to generate standard curve for 1301 Cell line. We used different wells every time to check if there was any significant variation. In addition, RealTime-PCR machine was calibrated before the start of these experiments by the company designated service engineer. Similarly, all pipettes to be used in DNA analysis were calibrated. To further minimize experimental variation, same individual conducted TL assay.   

Kotikalapudi R, Patel RK (2015), Comparative study of the influence of EDTA and sodium heparin on long term storage of cattle DNA. Cell J;17(1):181-6.

 Response: N/A (see above as DNA was isolated from blood collected in EDTA tubes.

  1. Lines 287-290 states "OD ratio (1.8-2.0) of 260:280 was used an indicator of DNA purity. Any sample where OD ratio was not in acceptable range, was re-isolated before DNA quantification. This information has been added in the relevant section of the revised manuscript." I assume the OD ratio was performed on the genomic DNA that was already extracted. Please include how many blood samples were needing to be thawed for DNA to be re-extracted again?

Response: DNA was collected from fresh blood samples collected on the day each participant visited the clinic. There were <1.0% samples where DNA has to be re-isolated as OD ratio was not within the acceptable range.

  1. I thank the authors for answering the question about using their standard curve derived from the 1301 cell line DNA to convert the T/S ratio into telomere length in base pair (bp) values. However, this method does not provide absolute base pair lengths in bp values; but rather mean relative telomere length measures. Please include how this method measures absolute telomere lengths in base pair values.

Response: We used DNA isolated from IMR90 cell line (human fibroblast primary cell line; cells collected after at different population doubling time: 10 time points of ~ 35 hours). Southern blot method (TRF) was used to determine telomere length. These values were then used to convert T/S ratio obtained in qPCR assay to base pair. The formula derived was later on used to convert T/S ratios to absolute telomere length: Absolute TL(bp) = 2433.23X + 3109.51 where X = T/S ratio of samples, 2433.23 is the slope and 3109.51 is the intercept) [72]. This information is now included in the methods section.

Wojcicki JM, Heyman MB, Elwan D, Shiboski S, Lin J, Blackburn E, Epel E (2015) Telomere length is associated with oppositional defiant behavior and maternal clinical depression in Latino preschool children. Trans Psychiatry 5:e581.    

To generate standard curve and converting T/S ratio into absolute TL (bp), DNA isolated from IMR90 cell line was used. DNA was isolated from cells taken from cultured cell line at different population doubling time (~35 hours; 10 time points). Southern blot method (TRF) was used to determine TL of these DNAs isolated from 10 different time points of sub-culturing. DNA isolated from these cells was also used in qPCR assay to determine T/S ratios as a part to generate initial standard curve which were then converted in to base pairs. The formula generated from this was later on used to convert T/S ratios to absolute telomere length: Absolute TL(bp) = 2433.23X + 3109.51 where X = T/S ratio of samples, 2433.23 is the slope and 3109.51 is the intercept [72].

  1. If using DNA extracted from the 1301 cell line of known length as the positive control for the standard curve, please include what measures are in place for storing this DNA to avoid telomere length attrition/degradation?

Response: DNA isolated from 1301 cell line and samples was stored at -80°C until required.

  1. Lines 305-308 states "The standard curve was then used to convert the T/S ratio into telomere length (TL) in base pairs (bp) using the following equation: Absolute 306 TL(bp) = 2433.23X + 3109.51 where X = T/S ratio, 2433.23 is the slope and 3109.51 is the intercept of the standard curve [71]." This reference does not mentioned using telomere length nor this equation. Please update this reference.

Response: We have also included the appropriate reference in the revised methods section:

Wojcicki JM, Heyman MB, Elwan D, Shiboski S, Lin J, Blackburn E, Epel E (2015) Telomere length is associated with oppositional defiant behavior and maternal clinical depression in Latino preschool children. Trans Psychiatry 5:e581. 

  1. I also thank the authors for including the M:F sample numbers in the figures. I believe the only difference in Figure 5 is on the Y axis with the sleep code (Figure 5A) and the telomere length (Figure 5B) however I notice a substantial difference in the M:F on the X axis which still add up to 136 females and 36 males.  Please include why some samples jump from being L-Hcy to H-Hcy or being L Mg to H Mg when measuring telomere length from sleep code as shown in Figures 5A and 5B?  This may account for some of the statistical significance found in these two figures.

Response: The numbers appearing on Fig 5A are actually wrong as we mistakenly used a wrong 2Way ANOVA file. We have corrected these numbers as these are same as shown on Fig 5B. We appreciated the reviewer for pointing out this mistake and we own this mistake.

  1. Similarly, please include how Figure 4c now has 144 females and 36 males listed in this figure?

Response: The numbers shown in Figure 4C (4th bar) is a typographical error. We have corrected this in the revised figure 4C. 

Round 3

Reviewer 3 Report

The authors have addressed all my concerns.  Thank you.

Author Response

We thank the reviewer 3 for all the constructive comments and for the final approval on the changes made